# Improving Sleep Quality, Daytime Sleepiness, and Cognitive Function in Patients with Dementia by Therapeutic Exercise and NESA Neuromodulation: A Multicenter Clinical Trial

**DOI:** 10.3390/ijerph20217027

**Published:** 2023-11-06

**Authors:** Esther Teruel-Hernández, José Antonio López-Pina, Sonia Souto-Camba, Aníbal Báez-Suárez, Raquel Medina-Ramírez, Antonia Gómez-Conesa

**Affiliations:** 1International School of Doctoral Studies, University of Murcia, 30100 Murcia, Spain; 2Faculty of Psychology, University of Murcia, 30100 Murcia, Spain; jlpina@um.es; 3Department of Physiotherapy, Medicine and Biomedical Sciences, University of A Coruña, 15006 A Coruña, Spain; sonia.souto@udc.es; 4Health Science Faculty, University of Las Palmas de Gran Canaria, 35016 Las Palmas, Spain; anibal.baez@ulpgc.es; 5SocDig Research Group, University of Las Palmas de Gran Canaria, 35016 Las Palmas, Spain; raquel.medina@ulpgc.es; 6Research Methods and Evaluation in the Social Sciences Research Group, Mare Nostrum Campus of International Excellence, University of Murcia, 30100 Murcia, Spain; agomez@um.es

**Keywords:** dementia, new technologies, physiotherapy, sleep quality, physical activity, neuromodulation

## Abstract

Dementia is a progressive decline in cognitive functions caused by an alteration in the pattern of neural network connections. There is an inability to create new neuronal connections, producing behavioral disorders. The most evident alteration in patients with neurodegenerative diseases is the alteration of sleep–wake behavior. The aim of this study was to test the effect of two non-pharmacological interventions, therapeutic exercise (TE) and non-invasive neuromodulation through the NESA device (NN) on sleep quality, daytime sleepiness, and cognitive function of 30 patients diagnosed with dementia (non-invasive neuromodulation experimental group (NNG): mean ± SD, age: 71.6 ± 7.43 years; therapeutic exercise experimental group (TEG) 75.2 ± 8.63 years; control group (CG) 80.9 ± 4.53 years). The variables were evaluated by means of the Pittsburg Index (PSQI), the Epworth Sleepiness Scale (ESS), and the Mini-Cognitive Exam Test at four different times during the study: at baseline, after 2 months (after completion of the NNG), after 5 months (after completion of the TEG), and after 7 months (after 2 months of follow-up). Participants in the NNG and TEG presented significant improvements with respect to the CG, and in addition, the NNG generated greater relevant changes in the three variables with respect to the TEG (sleep quality (*p* = 0.972), daytime sleepiness (*p* = 0.026), and cognitive function (*p* = 0.127)). In conclusion, with greater effects in the NNG, both treatments were effective to improve daytime sleepiness, sleep quality, and cognitive function in the dementia population.

## 1. Introduction

Dementia is a general term to describe a clinical neurodegenerative syndrome characterized by neuronal and synaptic loss, forming a brain deposit of intra- and/or extracellular insoluble protein aggregates [1,2]. In this disease, different domains are usually affected, such as memory, language, executive functions, behavior, and conduct. Sleep disorders, including reduced nighttime sleep time, sleep fragmentation, nocturnal wandering, and daytime sleepiness, among others, are very common [3,4].

A high prevalence of sleep disorders in patients with dementia has been identified for more than 30 years. Scientific evidence [5,6,7] estimates that between 25% and 66% of patients with dementia have a lower sleep efficiency and, consequently, fragmented sleep. This is considered one of the main causes of institutionalization [8].

Sleep is a basic biological need for the proper functioning of the organism and is essential for memory consolidation [9,10]. One of the mechanisms that coordinate sleep are the circadian rhythms, which are natural processes through physical, mental, and behavioral changes that follow a cycle of approximately 24.5–25 h. The main center of regulation is the suprachiasmatic nucleus (SCN) located in the hypothalamus, which acts as an endogenous clock in the sleep–wake cycle. When sunlight activates the SCN, it projects to adjacent areas of the hypothalamus that are related to temperature and circadian rhythms, and to the pineal gland [11,12]. As the usual sleeping hours approach, the stimulus of the SCN and thus the circadian activity decreases, increasing the homeostatic need for sleep. This is when, due to environmental darkness, endogenous melatonin (MLT) is synthesized, a hormone secreted by the pineal gland that regulates the wake–sleep cycle [9,10,11,12,13].

An alteration of the circadian system has an impact on MLT secretion, and this leads to a poor synchronization of biological rhythms such as the sleep–wake cycle. In people with dementia, the suprachiasmatic nucleus (SCN) is impaired and, therefore, this leads to a reduction in melatonin production, causing a disruption of the circadian rhythms, and, therefore, in the quality of life [12,13].

Currently, pharmacological options in dementia should be used with caution, seriously considering possible side effects before prescribing hypnotic and psychotropic agents [14]. When drug therapy is used, short-term use is recommended, since the occurrence of serious adverse effects and the lack of evidence on their chronic use are limitations for the person with dementia, given the absence of quality studies conducted in this population [14,15,16,17,18].

As for non-pharmacological strategies, there is currently a paucity of research in people with dementia, but even so, they are emerging as alternative procedures to improve sleep disorders in patients with dementia because of their minimal risk of side effects. These include sleep hygiene measures, light therapy, physical activity, cognitive stimulation, and auditory stimulation [19,20,21,22,23,24,25,26,27,28].

The need arises for effective non-pharmacological treatments backed by scientific evidence to support their use for the cognitive function and sleep disturbances suffered by this population.

In this sense, given that previous studies have shown that TE has demonstrated benefits on these characteristics, albeit in other neurodegenerative diseases [29,30,31], but its effect has not been studied in this population, it is of interest to check its possible benefit in this type of patient.

Similarly, some studies have shown positive evidence in the improvement of ANS-related dysfunctions in the field of multiple sclerosis [32] and cerebral palsy [33].

Therefore, the main objective of this study was to test the effect of two non-pharmacological interventions, on the one hand, therapeutic exercise (TE), and, on the other hand, non-invasive neuromodulation through the NESA device (NN), on sleep quality, daytime sleepiness, and cognitive function in patients with dementia.

## 2. Materials and Methods

### 2.1. Subjects

The sample of this study consists of 30 patients diagnosed with dementia who belong to two associations of Alzheimer’s and other dementias, where they perform daily classes of 1 h of physiotherapy for elderly and cognitive stimulation 5 days a week. During the study, patients in the different groups continued to receive these therapies. The new variation was the introduction of the TE and NN protocols. The inclusion criteria were obtaining a medical diagnosis of dementia equal to or greater than mild according to the Reisberg Global Deterioration Scale (GDS) [34], having stable medical and pharmacological conditions, as well as the ability to perform physical activity and follow verbal instructions. Also, patients were excluded if they had contraindications for the experimental treatments, such as: pacemakers, internal bleeding, ulcerated skin, acute febrile processes, cancer diagnosis, phobia of electricity, or comorbidity affecting sleep. This was in addition to those patients who were receiving drugs that interfere with sleep and acted as confounding factors. At the same time, the patients had the right of withdrawal; the voluntary decision of the patients or their caregivers to withdraw from the study at any time during the study, as well as any complication that might occur during the duration of the intervention, were considered grounds for withdrawal. The recruitment procedure was carried out by non-probabilistic convenience sampling.

### 2.2. Study Design

A randomized, single-blind, multicenter clinical trial was conducted in two associations of Alzheimer’s and other dementias to evaluate the effect of two non-pharmacological treatments, TE and NN, on sleep quality, daytime sleepiness, and cognitive function in patients with dementia. For this purpose, participants were randomly assigned to one of the three study groups (TEG; NNG; CG), using a fixed-size block design generated by the data manager to ensure a balanced randomization for each of the groups and in each of the participating centers. The allocation process was performed using probability convenience sampling [35]. The variables studied were collected at 4 different times during the study: at baseline, after 2 months (after completion of the NNG treatment), after 5 months (after completion of the TEG treatment), and after 7 months (after 2 months of follow-up). The specific process is shown in Figure 1.

### 2.3. Procedures

A randomized, multicenter trial was conducted to compare the treatment of NNG and TEG with a CG, and at the same time, both experimental treatments. The 10 participants of the CG received recommendations about sleep habits through an information leaflet but did not perform any active treatment [36,37].

### 2.4. Therapeutic Exercise Protocol

The 10 TEG participants received 52 sessions, from 10:00 am to 11:00 am, of an adapted cardiovascular exercise program in a small group format supervised by a physiotherapist. For the first 16 weeks, 3 weekly one-hour sessions were performed. Then, up to week 20, 1 session per week was followed by a progressive decrease in the load. The structure of the sessions was as follows: 1. active warm-up, 2. strength exercises, 3. balance and coordination exercises, 4. aerobic exercises, and 5. relaxation and return to calm. The sessions were gradually increasing in volume and intensity to achieve moderate intensity exercise. In addition, the participants’ caregivers were instructed to have the patients walk every day, gradually increasing their time, until they reached 30 min daily [38,39,40].

### 2.5. Nesa Non-Invasive Neuromodulation Protocol

The 10 NNG participants completed the protocol NESA [41,42,43,44,45,46,47] of 20 sessions of 60 min, three times a week. Each participant always had the same time for their session.

The non-invasive neuromodulation technique, NESA, is a non-invasive and easily transportable monitoring device, which emits low frequency microcurrents (1.3–14.28 Hz, depending on the program), low intensity (0.1–0.9 mA), and low voltage (±3 V) that are introduced into the distal nerve endings of the limbs by means of 24 electrodes (6 electrodes per limb, distributed between both wrists and ankles), producing a circulating bioelectric circuit in the body, for an estimated time to stimulate the autonomic nervous systems and enhance the recovery of those dysfunctional processes of the patient. In our case, dementia, it is known that there is a desynchrony in the wake–sleep cycle due to neurodegeneration, therefore, there is an alteration in the physiological processes of the circadian rhythm, this being a dysfunction in the segregation of melatonin, produced by the pineal gland [32,48,49]. For this, the treatment was performed in a centralized way to cover the nervous system in a general way, focusing the directional electrode in C7, and intensity “Low 3V” to favor the hormesis of the treatment. This location is close to the central nervous system, vagus nerve, and peripheral nervous system.

The NN protocol, administered by physiotherapists working in their respective centers, consisted of the distribution of 4 phases: The first phase was to avoid adverse effects, 3 sessions with program 1 (P1) 30 min, program 7 (P7) 15 min, and program (P8) 15 min. The second phase was to influence neuronal repolarization, 2 sessions with program 5 (P5) 30 min and P7 another 30 min. The third phase was to introduce the following, P5 15 min, P7 30 min, and P8 15 min. Finally, the fourth phase was to improve sleep quality, 12 sessions with P7 for 45 min and P8 15 min.

The microcurrents emitted by the different programs used were symmetrical biphasic low frequency and limited intensity, and therefore imperceptible to the patient.

### 2.6. Recovery Measures

Sleep quality: This was evaluated using the Pittsburgh Sleep Quality Index (PSQI) [50]. It consists of 19 items that analyze 7 different sleep components (subjective sleep quality, sleep latency, sleep duration, sleep efficiency, sleep disturbances, sleep efficiency, use of sleep medications, and daytime dysfunction). Each item is scored from 0 to 3. The total scale score ranges from 0 to 21 points where the lower end represents good sleep quality, and the upper end represents poor sleep quality. Cronbach‘s alpha of 0.83 obtained in Buysse et al. [50] for the PSQI components indicates a high degree of internal homogeneity. Therefore, the clinical and clinical properties of the PSQI suggest its usefulness in both psychiatric clinical practice and research.

Daytime sleepiness: This was evaluated using the Epworth Sleepiness Scale (ESS) [51]. It estimates the probability (0—never; 1—few; 2—moderate; 3—many) of falling asleep in eight different situations. Depending on the total score, which can vary between 0 and 24, the degree of sleepiness is determined. The higher the score, the greater the likelihood of daytime sleepiness. As the Murray Johns et al. [52] study shows, factor and item analyses have shown that the ESS is a unitary scale with high internal consistency (Cronbach’s alpha = 0.80). Daytime sleepiness has a high test–retest reliability over a 5-month period in normal subjects (r = 0.822, n = 87, *p* < 0.001).

Cognitive function: This was evaluated by means of the Mini-Cognitive Examination Test (Lobo’s MEC) [53]. Several studies recommend its use due to its effectiveness for the evaluation and follow-up of cases in which there is suspicion of cognitive impairment, obtaining reliable results, obtaining a sensitivity for dementia between 76–100%, and specificity between 78–100% [54,55]. It consists of 5 cognitive areas: orientation (temporal and spatial), fixation memory, concentration and calculation, delayed recall, and language and construction. The maximum score that can be obtained in this test is 35 points. If the patient obtains less than 24 points it is considered that there is some type of cognitive impairment. In the study of Buiza et al. [56] where they use the Lobo MEC scale to assess the cognitive status of patients with dementia, the test showed high internal consistency (α = 88), and good test–retest (0.64–1.00; *p* < 0.01) and inter-rater (0.69–1.00; *p* < 0.01) reliability, both for the total score and for each of the items. 

### 2.7. Statistical Analysis

Measurement of the variables was performed in all study participants at 4 different times: at baseline, 2 months (after completion of the NNG), 5 months (after completion of the TEG, and 7 months (after 2 months of follow-up). Self-administered questionnaires were used, but since our trial involved patients diagnosed with dementia, the responses were made by the primary caregiver of each patient.

Categorical variables were summarized using percentages and relative frequencies. Equality of proportions of the categories was compared using Pearson’s chi-square statistic. In addition, a one-way ANOVA was performed on the three groups to test whether differences were found as a function of age, and Levene’s test was used to test the homogeneity of variances of the groups.

The numerical variables were summarized using descriptive statistics (means, standard deviations pretest, at 2 months, at 5 months, and at 7 months). Since their distributions did not follow a normal law and the sample size was low, we chose to compare the three groups with the nonparametric Kruskal–Wallis H test. If the χ2 statistic was significant, the two-by-two comparison between the groups was performed with the Dwass–Steel–Critchlow–Fligner test to analyze the equality between the groups at each time point. The significance level for all analyses was set at *p* < 0.05. However, given the low sample size, the effect size (ES) was also calculated with Cohen’s typed mean difference. A low effect size was obtained when ES = 0.20, medium when ES = 0.50, and high when ES = 0.80. Statistical analyses were performed with SPSS v. 28.0 and JAMOVI v. 2.3.13. 

### 2.8. Ethics

As this was a sample of patients with dementia, their caregivers gave written informed consent before being assigned to a group and evaluated, and the rights of all participants were protected. All experimental protocols respected the fundamental principles established in the 1975 Declaration of Helsinki [57] and were approved by the Clinical Research Ethics Committee of the University of Murcia (registration number 3572/2021) and registered in ClinicalTrials.gov with identification number: NCT05715866.

## 3. Results

### 3.1. Sample

A total of 30 participants met the inclusion criteria and were randomized in the data collection process. Ten participants were assigned to the NNG, 10 to the TEG and 10 to the CG. The age of the participants was as follows: NNG 71.6 (SD = 7.43); TEG 75.2 (SD = 8.63), and CG 80.9 (SD = 4.53). (Table 1 shows the participants baseline clinical characteristics).

### 3.2. Effect of the Intervention on Sleep Quality

When comparing the three groups at each time point for sleep quality, using the Pittsburgh Sleep Quality Index (PSQI), significant differences were found after 5 months (*p* = 0.048) and after 7 months (*p* = 0.002). The NNG and TEG obtained improvements in sleep quality by decreasing both the scores of the test after 7 months; therefore, it was the NNG who showed a better evolution. In addition, the CG showed a worsening in sleep quality (Table 2 and Figure 2).

It was found that in the NNG, statistically significant differences were obtained at all measurement moments (*p* < 0.005) for sleep quality, except between months 2 and 5 (*p* = 0.129). Within the TEG, significant differences were obtained at all time points (Table 3).

### 3.3. Effect of Intervention on Daytime Sleepiness

When comparing the three groups at each time point of the variable, statistically significant differences were found at the time points (*p* < 0.001). The NNG and TEG obtained improvements in daytime somnolence by decreasing both the scores of the test after 7 months. Therefore, it was the NNG who showed a better evolution, and, in addition, the CG showed an increase in the test score, which led to a worsening in daytime sleepiness. (Table 4 and Figure 3).

Likewise, statistically significant differences in daytime sleepiness were found in the NNG at all measurement time points (*p* < 0.05), while in the TEG, significant differences were found between all the time points, except in the comparison between months 5 and 7 (*p* = 0.65) (Table 5).

### 3.4. Effect of Intervention on Cognitive Function

With respect to the cognitive function of the patients, statistically significant differences were found at the four measurement points. The NNG and TEG obtained improvements in cognitive function reaching 7 months with a score of 30.7 (SD = 3.50) in the NNG, and 27.5 (SD = 2.92) in the TEG. This means that, although both groups improved in the cognitive function scores, it was the NNG patients who showed a better evolution. With respect to the CG, the results showed a small worsening in cognitive function (Table 6 and Figure 4).

In the NNG, improvements in cognitive function were significant at all the time points (*p* < 0.005), improving patients by 35%. However, in TEG, no statistically significant differences in cognitive function were obtained at baseline compared to 2 months (*p* = 0.52), but significant differences were obtained at all the other measurement time points (*p* < 0.04) (Table 7).

## 4. Discussion

The main objective of this study was to test the effect of two non-pharmacological interventions on sleep quality, daytime sleepiness, and cognitive function in patients with dementia. The sample proved to be homogeneous, despite the clinical variability that may present in this type of population and it being a multicenter intervention.

In this regard, the results obtained with respect to the sleep quality of patients with dementia showed that the group treated with NESA noninvasive neuromodulation obtained an improvement of 3.3% over the therapeutic exercise group, even from similar values in the pretest with the therapeutic exercise group. Both treatments seemed to be effective for the improvement of sleep quality; however, the efficacy of the treatment in the NNG stands out, obtaining a lower score after 7 months than the TEG. In the study by Cao, S et al. [58] an improvement in the overall PSQI score of ≥3 points was considered a minimal clinically important difference. In our study, in the NNG group the score went from 20.6 to 10.5; in the TEG from 20.1 to 11.2; and in the CG from 20.8 to 20.6.

Similar results were obtained when measuring daytime sleepiness in patients with dementia, highlighting the score of the NNG, which obtained a lower score, and with an improvement of 36.7% over the baseline. The TEG also showed an improvement of 19.17% at the end of the study with respect to the baseline, however, with a smaller effect than the NNG. On the ESS scale, the minimum clinically important improvement in the ESS is estimated [59] to be between −2 and −3. Given this variance in the ESS, it is time to reconsider the MCID to be between −5 and −6. In our study, the NNG went from a score of 15.2 to 6.40; in the TEG from 14.7 to 10.1; and in the CG from 15.4 to 16.5.

Regarding the cognitive function of patients with dementia, the NNG modality obtained better results than the TEG and CG. The results showed that the no intervention group (CG) did not improve in cognitive function, and the TEG patients also obtained some improvements in the cognitive function of the patients; however, it seemed that the highest effect was in the NNG. In addition, J.S. Andrews et al. [60] involving a survey of neurologists and geriatricians reported a mean MCID for the scale of 3.75 (95% confidence interval: 3.5–3.95); in our study the NNG went from a score of 22.7 to 30.7, the TEG from 23.9 to 27.5, and the CG from 18.6 to 18.3.

Changes in sleep quality in people with dementia represent an important challenge for the scientific community since, for example, sleep and circadian rhythm disturbances are very common in patients with this pathology and up to 45% of patients experience sleep problems. The clinical presentation is characterized by memory loss and cognitive dysfunction [61], as well as increased health care costs and mortality. Current treatments include traditional pharmacological and non-pharmacological approaches, with limited efficacy [61,62,63,64]. Non-pharmacological interventions have been performed with mixed results, such as aromatherapy (without significant results on the PSQI) [65], acoustic stimulation [66], and transcranial stimulation [67]. Therefore, there is an urgent need to develop new alternative techniques to the existing ones.

Non-invasive brain stimulation is of great interest in this context [66,67]. In our study, we observed that 20 sessions of 60 min duration of noninvasive electrical neurostimulation (located in the hands and wrists without the need for cranial electrodes) with the NESA device generated greater relevant and lasting changes over time, in sleep disturbances and cognitive function in the patients with dementia who participated in this study.

This is the first study to use the NESA noninvasive neuromodulation with patients with dementia and yields similar results in sleep parameters to those obtained by Garcia et al. [68] who used the same device to cause noninvasive brain stimulation in basketball players. Regarding the use of noninvasive brain stimulation to facilitate sleep in patients with sleep disorders, it has been used with different modalities in small samples, with contradictory results. Thus, Jiang et al. [66] who used repetitive transcranial magnetic stimulation (rTMS) in the dorsolateral prefrontal cortex for 30 min/day, for 2 weeks in patients with chronic primary insomnia, achieved a greater increase in the duration of rapid eye movement (REM) sleep than with pharmacological treatments. In contrast, Saebipur et al. [67] found no effect on REM sleep duration using the rTMS technique in patients with insomnia. The strength of the device used in our study, NESA, lies in the use of microcurrents imperceptible to the patient and without polar effects that modulate the autonomic nervous system, obtaining benefits in sleep.

Where benefits in sleep quality like our study were also recorded were in those where the intervention was oriented towards education in sleep habits [69], or in the prescription of a physical exercise pattern [70]. These results are also consistent with the findings of previous experimental studies, in which an exercise modality like our study was observed with cognitive tasks/engagement in cognitively impaired older adults [71,72]. Perhaps the major difference with our study corresponds to the number of weeks of training (20 weeks vs. 12 weeks).

Although after the maintenance period, the result was below the level of minimum poor sleep quality; we believe that this could be improved if after further research following this line of treatment, a greater number of treatment sessions were done. Having found an improvement in sleep in both the NNG and TEG we consider that to be a great advance for this current problem. The next future goal should be to recover 100% sleep quality; although, since this is a neurodegenerative disease and there is a dysfunction in melatonin secretion, achieving optimal sleep quality could be a very complex goal [32,48,49]”.

Regarding physical exercise in people with dementia, according to the studies, it appears that systematic exercise, through various mechanisms, can promote brain function and maintain and improve both cognitive and physical functions [73]. Unfortunately, several previous studies do not mention the level of intensity, duration, and frequency of exercise needed for optimal exercise intervention in people with dementia [74,75]. Perhaps this is one of the strengths of our study, in which the effort has been made to describe in detail the dose of exercise prescribed.

Cognitive functions and their influence under noninvasive electrical stimulation in people with dementia have been the subject of study in recent years, reporting results as encouraging as in our study [76,77]. There are several techniques with variability of results, such as vagus nerve stimulation, deep brain stimulation (DBS) and anticonvulsant magnetic therapy (MST) [76,78,79,80]. One of the advantages in the use of NESA neuromodulation is based on the non-occurrence of secondary events, as opposed to the detriment of DBS which sometimes leads to the presence of hemorrhage, seizures, and infection, or of MST in which, on some occasions, leads to cases of discomfort caused by muscle spasms of the scalp or face, headaches, and seizures [72]. Therefore, NN could constitute a complementary alternative for cognitive rehabilitation treatment in the dementia population since the changes observed in the NNG have shown objective evidence of functional modifications in cognition from baseline.

One of the keys in the possible neurophysiological explanation that we could hypothesize in the improvement of the drowsiness state and cognitive level of our patients, focuses on the influence of the locus coeruleus (LC) [81], involved in many of the sympathetic effects during stress due to increased production of noradrenaline, as well as being a key center in the processes of wakefulness. In patients with dementia there is significant atrophy within the LC, which is the reason for neuronal and noradrenaline loss. With this reasoning, different studies show a direct influence of non-invasive neuromodulation on the LC, both in animals [82,83,84] and in human studies [83,84,85,86,87].

Given that this is the first preliminary study in the world using the NESA noninvasive neuromodulation device in a dementia population with the aim of improving sleep quality, daytime sleepiness, and cognitive function, it would be desirable to conduct further studies using other more objective measurements to corroborate these satisfactory results. In this sense, analyzing sleep–wake parameters by means of an actigraphy could be a reliable and more precise indicator to evaluate the different phases of sleep and the changes during the whole cycle, obtaining additional information on the total duration of sleep, actual time of falling asleep, patient’s sleep conditions, etc. [88,89]. On the other hand, and given the importance for sleep quality in evaluating the change between the sympathetic and parasympathetic balance of the ANS during the day, it could be useful to measure the heart rate variability by means of an electrocardiogram, recording the electrical conduction system, and myocardial contraction [90,91,92,93].

This study begins an interesting field of research in the neuromodulation of the autonomic system, sleep, and cognitive function as a daily part of the recovery in patients with dementia.

### Limitations

The study had some limitations that we recommend solving in future interventions. The study had a small sample size and was from a specific area, so the generalizability of the results is limited. Future research should be conducted in a variety of settings with larger samples to determine these measurement pathways in more detail. Since some results were based on self-reports (PSQI, MEC) from face-to-face interviews, the recording model should be further examined using other measures such as actigraphy or polysomnography, although it is known that due to the cognitive limitations of patients these techniques are complicated to perform. The variables included do not explain all the possible variance, so other possible variables may also mediate the relationship, such as fatigue, self-efficacy, stress, or even the influence of their social and/or family environment.

## 5. Conclusions

In conclusion, two non-pharmacological treatments, therapeutic exercise, and non-invasive neuromodulation NESA, appear to be effective treatments to improve daytime sleepiness, sleep quality, and cognitive function.

## Figures and Tables

**Figure 1 ijerph-20-07027-f001:**
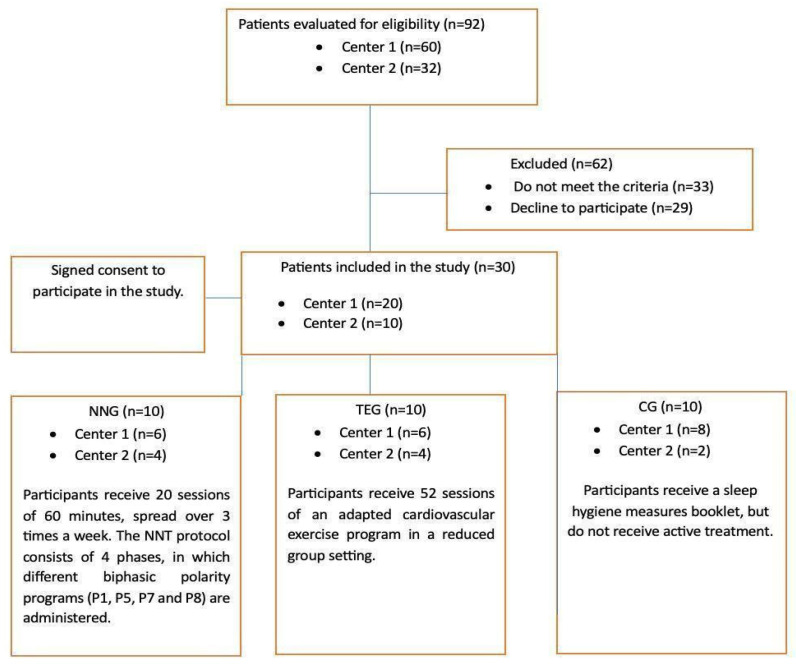
Study design flowchart.

**Figure 2 ijerph-20-07027-f002:**
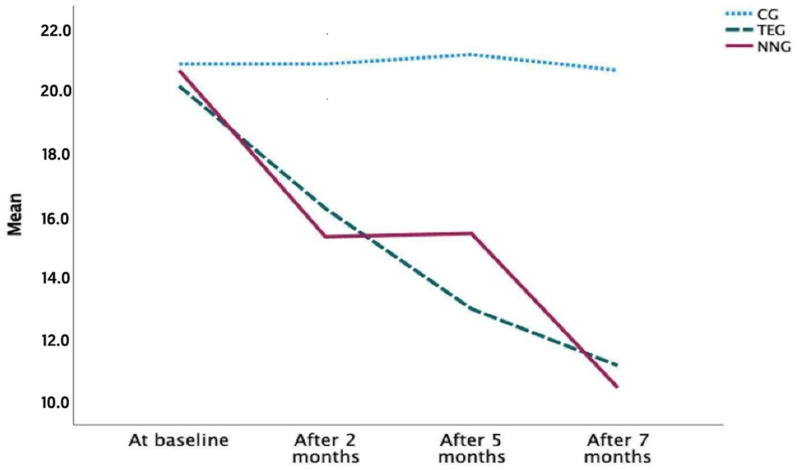
Evolution of sleep quality in different periods.

**Figure 3 ijerph-20-07027-f003:**
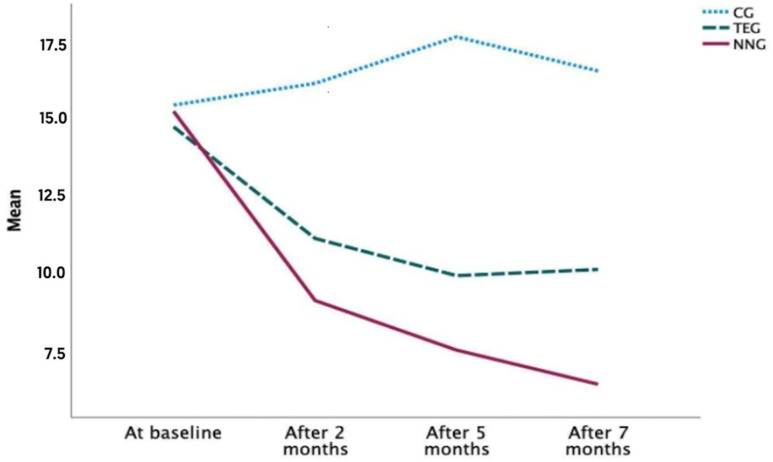
Evolution of daytime sleepiness in different periods.

**Figure 4 ijerph-20-07027-f004:**
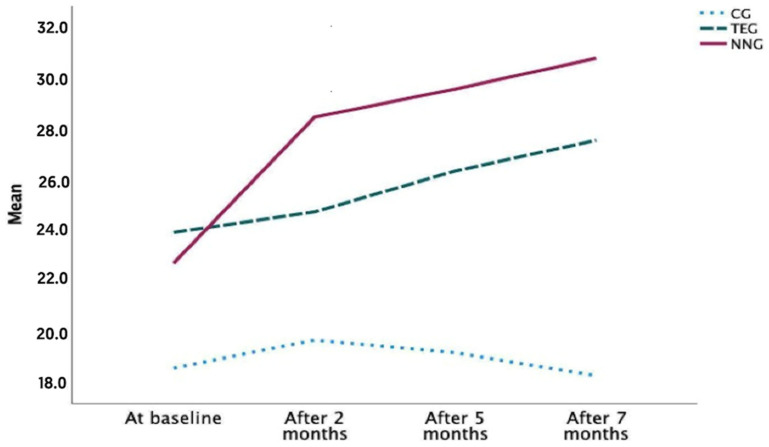
Evolution of cognitive function in different periods.

**Table 1 ijerph-20-07027-t001:** Participant baseline clinical characteristics.

Variable	Option	Neuromodulation No Invasive Group (N = 10)	Therapeutic Exercise Group (N = 10)	Control Group (N = 10)	*p*-Value
Gender	Female	6	4	8	0.189
	Male	4	6	2	
Insomnia	No	4	5	5	0.875
	Yes	6	5	5	
Daytime sleepiness	No	6	4	5	0.670
	Yes	4	6	5	
SAOS	No	7	7	7	1.000
	Yes	3	3	3	
Parasomnias	No	4	8	5	0.171
	Yes	6	2	5	
Snoring	No	2	2	3	0.83
	Yes	8	8	7	
Type of dementia	Alzheimer	8	8	9	0.666
	Lewy bodies	1	1	1	
	Parkinson	1	0	0	
Daytime walks	No	4	7	4	0.301
	Yes	6	3	6	
Sedentary life	No	7	6	8	0.621
	Yes	3	4	2	
Rheumatic disease	No	8	10	7	0.186
	Yes	2	0	3	
Symptoms of gastroesophageal	No	9	9	10	0.585
	Yes	1	1	0	
Prostate disease	No	8	8	9	0.787
	Yes	2	2	1	
Cardiomyopathy	No	8	9	10	0.329
	Yes	2	1	0	
Depression	No	3	7	6	0.175
	Yes	7	3	4	
Anxiety	No	6	10	7	0.089
	Yes	4	0	3	
Parkinson’s disease	No	9	10	9	0.585
	Yes	1	0	1	
Stroke	No	9	10	7	0.133
	Yes	1	0	3	
Treatment with bronchodilators	No	9	9	9	1.00
	Yes	1	1	1	
Treatment with thyroxine	No	9	9	6	0.153
	Yes	1	1	4	
Treatment with diuretics	No	9	8	6	0.271
	Yes	1	2	4	
Treatment with antidepressants	No	3	6	3	0.287
	Yes	7	4	7	
Treatment with neuroleptics	No	10	8	6	0.082
	Yes	0	2	4	
Treatment with benzodiazepines	No	7	7	8	0.843
	Yes	3	3	2	

**Table 2 ijerph-20-07027-t002:** Means, standard deviations, effect size, and statistical significance for intergroup and intragroup comparison of sleep quality in the different periods.

PSQI	*p*-Value	NNG Mean (SD)	TEG Mean (SD)	CG Mean (SD)	Pairwise Comparison	d	*p*-Value
At baseline	0.664	20.6 (8.32)	20.1 (8.75)	20.8 (6.30)	NNG–TEG NNG–CG TEG–CG		
After 2 months	0.060	15.3 (5.50)	16.2 (9.07)	20.8 (6.11)	NNG–TEG NNG–CG TEG–CG	−0.12 −0.95 −0.60	0.980 0.141 0.102
After 5 months	0.048	15.4 (6.47)	13.0 (7.04)	20.1 (6.44)	NNG–TEG NNG–CG TEG–CG	0.36 −0.88 −1.20	0.609 0.203 0.059
After 7 months	0.002	10.5 (5.13)	11.2 (6.00)	20.6 (6.90)	NNG–TEG NNG–CG TEG–CG	−0.13 −1.99 −1.76	0.972 0.004 0.009

**Table 3 ijerph-20-07027-t003:** Two-to-two within-group comparisons of sleep quality at different time points.

PSQI	Moments	x^2^	*p*-Value	d
NNG	At baseline–After 2 months	5.37	<0.001	1.556
	At baseline–After 5 months	6.93	<0.001	1.150
	At baseline–After 7 months	12.75	<0.001	2.328
	After 2 months–After 5 months	1.57	0.129	−0.019
	After 2 months–After 7 months	7.38	<0.001	1.384
	After 5 months–After 7 months	5.81	<0.001	1.099
TEG	At baseline–After 2 months	6.09	<0.001	1.813
	At baseline–After 5 months	11.63	<0.001	1.994
	At baseline–After 7 months	15.51	<0.001	2.846
	After 2 months–After 5 months	5.54	<0.001	0.863
	After 2 months–After 7 months	9.42	<0.001	1.405
	After 5 months–After 7 months	3.88	<0.001	0.818
CG	At baseline–After 2 months	0.000	1.000	0.000
	At baseline–After 5 months	0.957	0.347	−0.098
	At baseline–After 7 months	2.871	0.008	−0.422
	After 2 months–After 5 months	0.957	0.347	−0.191
	After 2 months–After 7 months	2.871	0.008	−0.767
	After 5 months–After 7 months	1.914	0.066	−0.843

**Table 4 ijerph-20-07027-t004:** Means, standard deviations, effect size, and statistical significance for the intergroup and intragroup comparison of daytime sleepiness in the different periods.

ESE	*p*-Value	NNG Mean (SD)	TEG Mean (SD)	CG Mean (SD)	Pairwise Comparison	d	*p*-Value
At baseline	0.863	15.2 (2.39)	14.7 (1.83)	15.4 (3.13)	NNG–TEG NNG–CG TEG–CG		
After 2 months	<0.001	9.10 (2.13)	11.1 (2.64)	16.1 (3.51)	NNG–TEG NNG–CG TEG–CG	0.83 2.41 1.61	0.132 <0.001 0.007
After 5 months	<0.001	7.50 (3.44)	9.90 (3.21)	17.6 (2.84)	NNG–TEG NNG–CG TEG–CG	0.72 3.20 2.54	0.278 <0.001 <0.001
After 7 months	<0.001	6.40 (3.10)	10.1 (2.60)	16.5 (2.80)	NNG–TEG NNG–CG TEG–CG	1.29 3.42 2.37	0.026 <0.001 <0.001

**Table 5 ijerph-20-07027-t005:** Two-to-two intragroup comparisons in daytime sleepiness at the different time points.

ESE	Moments	x^2^	*p*-Value	d
NNG	At baseline–After 2 months	4.75	<0.001	1.707
	At baseline–After 5 months	6.84	<0.001	1.579
	At baseline–After 7 months	9.69	<0.001	2.052
	After 2 months–After 5 months	2.09	0.046	0.663
	After 2 months–After 7 months	4.94	<0.001	1.045
	After 5 months–After 7 months	2.85	0.008	0.803
TEG	At baseline–After 2 months	4.084	<0.001	1.62
	At baseline–After 5 months	6.958	<0.001	1.656
	At baseline–After 7 months	6.504	<0.001	1.776
	After 2 months–After 5 months	2.874	0.008	0.775
	After 2 months–After 7 months	2.420	0.023	0.567
	After 5 months–After 7 months	0.454	0.654	−0.124
CG	At baseline–After 2 months	2.027	0.053	−0.661
	At baseline–After 5 months	6.081	<0.001	−1.938
	At baseline–After 7 months	2.896	0.007	−0.919
	After 2 months–After 5 months	4.054	<0.001	−1.273
	After 2 months–After 7 months	0.869	0.393	−0.280
	After 5 months–After 7 months	3.185	0.004	1.256

**Table 6 ijerph-20-07027-t006:** Means, standard deviations, effect size, and statistical significance for intergroup and intragroup comparison in cognitive function in the different periods.

MEC de Lobo	*p*-Value	NNG Mean (SD)	TEG Mean (SD)	CG Mean (SD)	Pairwise Comparison	d	*p*-Value
At baseline	0.020	22.7 (3.27)	23.9 (3.60)	18.6 (5.10)	NNG–TEG NNG–CG TEG–CG		
After 2 months	0.002	28.4 (4.48)	24.7 (3.13)	19.7 (4.83)	NNG–TEG NNG–CG TEG–CG	−0.96 −1.87 −1.23	0.231 0.005 0.044
After 5 months	<0.001	29.5 (4.01)	26.3 (2.83)	19.2 (4.87)	NNG–TEG NNG–CG TEG–CG	−0.92 −2.31 −1.78	0.162 0.001 0.010
After 7 months	<0.001	30.7 (3.50)	27.5 (2.92)	18.3 (4.27)	NNG–TEG NNG–CG TEG–CG	−0.99 −3.18 −2.52	0.127 <0.001 <0.001

**Table 7 ijerph-20-07027-t007:** Two-to-two intragroup comparisons in cognitive function at the different time points.

MEC de Lobo	Moments	x^2^	*p*-Value	d
NNG	At baseline–After 2 months	6.22	<0.001	−2.519
	At baseline–After 5 months	10.20	<0.001	−3.750
	At baseline–After 7 months	13.44	<0.001	−6.000
	After 2 months–After 5 months	3.98	<0.001	−1.256
	After 2 months–After 7 months	7.22	<0.001	−1.302
	After 5 months–After 7 months	3.24	0.003	−0.976
TEG	At baseline–After 2 months	0.649	0.522	−0.402
	At baseline–After 5 months	3.244	0.003	−1.057
	At baseline–After 7 months	5.449	<0.001	−1.390
	After 2 months–After 5 months	2.595	0.015	−1.488
	After 2 months–After 7 months	4.800	<0.001	−1.729
	After 5 months–After 7 months	2.206	0.036	−1.162
CG	At baseline–After 2 months	1.911	0.067	−0.575
	At baseline–After 5 months	0.597	0.555	−0.254
	At baseline–After 7 months	1.553	0.132	0.146
	After 2 months–After 5 months	1.314	0.200	0.514
	After 2 months–After 7 months	3.464	0.002	1.302
	After 5 months–After 7 months	2.150	0.041	0.752

## Data Availability

The raw data supporting the conclusions of this article will be made available by the authors without undue reservation.

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
