# Peer review of "Improving Sleep Quality, Daytime Sleepiness, and Cognitive Function in Patients with Dementia by Therapeutic Exercise and NESA Neuromodulation: A Multicenter Clinical Trial"

_ijerph, 2023, doi:10.3390/ijerph20217027_

Round 1

Reviewer 1 Report

Comments and Suggestions for Authors

Thank you for reviewing the paper titled “Improving sleep quality, daytime sleepiness, and cognitive 3 function in patients with dementia by therapeutic exercise and 4 NESA neuromodulation. Multicenter Clinical Trial.” The paper is organized, well-written, but some question raised into consideration:

Introduction:  need more example on studies which is similar or against to this one.

ABSTRACT: very good, but the method needs to be mentioned clearly.

“where they perform daily classes 80 of 1h of physiotherapy for elderly and cognitive stimulation 5 days a weekMention exclusion criteria for the study please” .. Why this period of follow up .is there is a reference provided in previous papers?

29 participant decline to continue, please mention the reason.

These programs use a biphasic polarity. The patient does not perceive any physical 149 sensation of current due to the parameters used.I cannot understand this sentence.

ANOVA, please mention one or two way ANOVA.

Discussion: unfortunately, does not cover all results

Conclusion needs to be more clear and obvious.

Some sentences of conclusion should be better addressed in the discussion section.

Check the manuscript English language and grammar

Check references well

Comments on the Quality of English Language

Check the manuscript English language and grammar

Author Response

1. Introduction:  need more example on studies which is similar or against to this one.
Response: Thank you for your contributions to improving the manuscript.. To clarify, we have included in the introduction section (page 2, lines 75-83) the following text : “The need arises for effective non-pharmacological treatments backed by scientific evidence to support their use for the cognitive function and sleep disturbances suffered by this population. In this sense, given that previous studies have shown that ET has demonstrated benefits on these characteristics, albeit in other neurodegenerative diseases [29-31], but its effect has not been studied in this population, it is of interest to check its possible benefit in this type of patient. Similarly, some studies have shown positive evidence in the improvement of ANS-related dysfunctions in the field of multiple sclerosis [32] and cerebral palsy [33]".
2. Abstract: very good, but the method needs to be mentioned clearly.
Response: Thank you for this contribution. In order to better understand the method, a more extensive and detailed explanation has been added to the Abstract’s section (page 1, lines 18-26): “The aim of this study is to test the effect of two non-pharmacological interventions, therapeutic exercise (TE) and non-invasive neuromodulation through the NESA device (NN) on sleep quality, daytime sleepiness, and cognitive function of 30 patients diagnosed with dementia [non-invasive neuromodulation experimental group (NNG): mean ±SD, age: 71.6 ± 7.43 years; therapeutic exercise experimental group (TEG) 75.2± 8.63 years, control group (CG) 80.9±4.53 years]. The aforementioned variables were evaluated by means of the Pittsburg Index (PSQI), the Epworth Sleepiness Scale (ESS)  and the Mini-Cognitive Exam Test  at 4 different times during the study: at baseline, after 2 months (after completion of the NNG), after 5 months (after completion of the TEG) and after 7 months (after 2 months of follow-up)”
3. “where they perform daily classes of 1h of physiotherapy for elderly and cognitive stimulation 5 days a week. Mention exclusion criteria for the study please .
Response: Thank you for your input. The exclusion criteria are indicated in section "2.1 Subjects" as follows: “Also, patients were excluded if they had contraindications for the experimental treatments, such as: pacemakers, internal bleeding, ulcerated skin, acute febrile processes, cancer diagnosis, phobia of electricity, comorbidity affecting sleep. In addition to those patients who were receiving drugs that interfere with sleep and act as confounding factors. At the same time, the patients had the right of withdrawal; the voluntary decision of the patients or their caregivers to withdraw from the study at any time during the study, as well as any complication that might occur during the duration of the intervention, were considered grounds for withdrawal”
4. Why this period of follow up .is there is a reference provided in previous papers?
Response: Thank you for this question. Prior to the implementation of this study, a systematic review was carried out to find out which treatments are currently the most commonly used to treat these disorders in the dementia population. As shown in the introduction (page 2, lines 69-73): “As for non-pharmacological strategies, there is currently a paucity of research in people with dementia, but even so, they are emerging as alternative procedures to improve sleep disorders in patients with dementia because of their minimal risk of side effects. These include sleep hygiene measures, light therapy, physical activity, cognitive stimulation, and auditory stimulation. [19-28]“. Since the analysis of these studies revealed little clarity on their positive findings and the follow-up period of these studies was between 2 weeks and 1 month, it was decided to perform a slightly longer follow-up period (2 months follow-up).  Some of the studies on which this decision was based are those mentioned in the literature, such as: Burns et al [19] and Dowling et al [20].
5. 29 participants decline to continue, please mention the reason.
Response: Thank you for your input. As shown in figure 1, of the 92 subjects selected, 29 refused to participate.  This refusal came from the caregivers, who after being informed of the study procedures in detail, via the information sheet, did not sign the informed consent form and therefore these subjects could not be counted on for inclusion in the study. The reason for not signing the informed consent form is unknown.
6. “These programs use a biphasic polarity. The patient does not perceive any physical sensation of current due to the parameters used.”I cannot understand this sentence.
Response: Thank you for your input. This phrase refers to the non-invasive neuromodulation device NESA. It is a superficial treatment with micro-electrical current (generated by the device) that produces imperceptible sensations through low impedance zones (where the semi-electrodes corresponding to the exact location of the nerve are installed). In other words, once the electrodes have been placed, the device allows the electrical signal to enter, creating the possibility of modulating the nervous system through a current without polar effects. This electrical signal is a symmetrical biphasic microcurrent, of low frequency and limited intensity (depending on the programme chosen), and therefore imperceptible to the patient.
For the reader's convenience, the sentence has been replaced (page 5, line 164) by: “The microcurrents emitted by the different programs used are symmetrical biphasic, low frequency and limited intensity, and therefore imperceptible to the patient.”
7. ANOVA, please mention one or two way ANOVA.
Response: Thank you for your input. We have replaced the sentence (page 6, line 204) with: "one-way ANOVA".
8. Discussion: unfortunately, does not cover all results
Response: Thank you for pointing this out. These have been added to the "discussion" section to reflect the value of our results on each variable (page 14, line 300): "In the study by Cao, S et al [58] an improvement in PSQI global score of ≥ 3 points was considered a minimal clinically important difference. In our study, in the NNG group the score goes from a score of 20.6 to 10.5; in TEG from 20.1 to 11.2; and in CG from 20.8 to 20.6."; (page 14, line 308): On the ESS scale, the minimum clinically important improvement in ESS is estimated [59] to be between -2 and -3. Given this variance in ESS, it is time to reconsider the MCID to be between -5 and -6. In our study, NNG goes from a score of 15.2 to 6.40; in TEG from 14.7 to 10.1; in CG from 15.4 to 16". 5. ; (page 15, line 317) : In addition, the study by J.S. Andrews et al.[60] involving a survey of neurologists and geriatricians reported a mean MCID for the scale of 3.75 (95% confidence interval 3.5-3.95), in our study NNG from a score of 22.7 to 30.7, TEG from 23.9 to 27.5 and CG from 18.6 to 18.3.
9. Conclusion needs to be more clear and obvious. Some sentences of conclusion should be better addressed in the discussion section.
Response: Thank you for your input. We have simplified the conclusion (page 17, lines 421-423 : “In conclusion, two non-pharmacological treatments, therapeutic exercise and non-invasive neuromodulation NESA appear to be effective treatments to improve daytime sleepiness, sleep quality and cognitive function”.
In addition, the discussion has been modified by highlighting in yellow what has been added in the "Discussion" section.
10. Check the manuscript English language and grammar
Response: Thank you for your input, we have checked the grammar.
11. Check references well
Response: Thank you for your input. We have added new references in the introduction section, and have therefore updated and checked the references.

Reviewer 2 Report

Comments and Suggestions for Authors

1 I would like to know the time of day (morning, afternoon, evening?) at which the treatment procedures were carried out

2 In my opinion the sample is very small to draw conclusions on dementia.

Maybe the treatment effects were different when depression, SAOS, and vascular pathology were excluded

Author Response

1. I would like to know the time of day (morning, afternoon, evening?) at which the treatment procedures were carried out.
Response: Thank you for this question. As shown on page 3 of the "Study design" section, the sessions were held at the participants' two centres, and therefore, the timetable for the sessions was during their opening hours. Therefore, the TEG was conducted in the morning from 10:00am-11:00a.m (before lunch), and the NNG rotated the device as one finished the session, starting at 08:30 am until 19:00p.m, but each participant always had the same schedule.  
For better understanding, the following sentences have been added to the text: (page 4, line 142) "Each participant always had the same time for their session", and "from 10:00 to 11:00" (page 4, line 131).
2. In my opinion the sample is very small to draw conclusions on dementia
Response: Thank you for this contribution. As discussed in the Limitations section (page 16, lines 409-412): The study has some limitations that we recommend to be addressed in future interventions. The study had a small sample size and came from a specific area, so the generalisability of the results is limited. Future research should be conducted in a variety of settings with larger samples to determine these measurement pathways in more detail.
3. Maybe the treatment effects were different when depression, SAOS, and vascular pathology were excluded.
Response: Thank you for this contribution. As shown on (page 3, lines 100-107), the inclusion criteria were marked as having stable medical and pharmacological conditions, as well as their ability to perform physical activity and follow verbal instructions, while also excluding those patients who had contraindications to the experimental treatments, including ulcerated skin and internal haemorrhage. Thus, although some patients, as shown in table 1 of baseline characteristics of the participants, had OSAHS, depression and vascular pathology, these were not of severe intensity. Nevertheless, it would be interesting to take this into account in future research following this line of treatment.

Reviewer 3 Report

Comments and Suggestions for Authors

The authors need to provide power calculations to justify the sizes of the groups.

It is not clear to me from the data how it can be said that the control group had worsening sleep quality over the seven months of the study.

In a randomized controlled trial with parallel groups, the comparisons within groups over time do not matter. the comparisons between groups are important. So Tables 3, 5 and 7 are unnecessary.

The people in the two treatment groups had much more contact with health care professionals than those in the control group. How much was this contact responsible for the benefits rather than the treatments themselves?

Is there an explanation why those in the treatment groups continued to improve after the treatments were finished?

What are the minimally important clinical differences in the Lobo MEC, ESS and PSQI? What proportions of subjects in each group had clinically important improvements in each of the parameters?

There is a misprint on page 4. Murray Johns developed the ESS, not J.M. Murray.

Author Response

1. The authors need to provide power calculations to justify the sizes of the groups.
Response: Thank you for this contribution. This information has been added in the subject section (page 3 line 109): Power reached is 0. 389: TE of 0.4 (medium-high); Equal errors: Beta= alpha;N= 10+10+10=30; Groups= 3; No. of measurements= 4
2. It is not clear to me from the data how it can be said that the control group had worsening sleep quality over the seven months of the study.
Response: Thank you for your input.Sleep quality was measured by the Pittsburgh Sleep Quality Index (PSQI), and as shown on page 5 line 167-174: The lower the score on the scale, the better the patient's sleep quality. Table 2 shows how, when comparing the 3 groups at each time point for the sleep quality variable, statistical differences were found at 5 months (p=0.048) and 7 months (p=0.002). The NN and ET groups have obtained improvements in sleep quality by decreasing both the test score after 7 months [10.5 (SD=5.13) and 11.2 (SD=6.00) respectively], but the CG showed an increase in the test score, which means a worsening in sleep quality [20.8 (SD=6.30) in the pretest stage and, 20.6 (SD=6.90) after 7 months].
3. In a randomized controlled trial with parallel groups, the comparisons within groups over time do not matter. The comparisons between groups are important. So Tables 3, 5 and 7 are unnecessary.
Response: Thank you for your contributions to improving the manuscript.  We believe that you are correct, however, given that this is an innovative approach, we wanted to detect and rule out possible biases, and to be more precise in drawing conclusions and confirming our hypotheses. 
4. The people in the two treatment groups had much more contact with health care professionals than those in the control group. How much was this contact responsible for the benefits rather than the treatments themselves?
Response: Thank you for this question. Both the treatment and control groups have contact with health professionals. This is because all participants continued to receive the therapies provided by their respective associations. The only variation was the introduction of the TE and NN protocol. Therefore, the control group continued to receive daily 1h classes of physiotherapy for the elderly and cognitive stimulation 5 days a week. 
This information can be seen in section 2.1 Sample (page 2, line 92-97) where the following is mentioned: "The sample of this study consists of 30 patients diagnosed with dementia who belong to two Alzheimer's and other dementia associations, where they receive daily classes of 1h of physiotherapy for the elderly and cognitive stimulation 5 days a week. During the study, patients in the different groups continued to receive these therapies. The new variation was the introduction of the TE and NN protocol".
5. Is there an explanation why those in the treatment groups continued to improve after the treatments were finished?
Response: Thank you for pointing this out. As shown in the "Study Design" section (page 3, lines 118-122):  “The variables studied were collected at 4 different times during the study: at baseline, after 2 months (after completion of the NNG treatment), after 5 months (after completion of the TEG treatment) and after 7 months (after 2 months of follow-up)”. This means that NNG (20 sessions, 3 times a week) lasted 2 months, and therefore, the evaluations "after 5 months" and "after 7 months" are considered as a follow-up period, obtaining favourable results for the variables. On the other hand, TEG (52 sessions, 3 times a week) lasted 5 months, and the evaluation "after 7 months" is considered a follow-up period of 2 months, thus obtaining favourable results for the study variables.
6. What are the minimally important clinical differences in the Lobo MEC, ESS and PSQI? What proportions of subjects in each group had clinically important improvements in each of the parameters?
Response: Thank you for pointing this out. In the study of Cao, S et al. [58], the difference in PSQI was calculated by subtracting the score at 6 months postoperative from the preoperative value. Additionally, an improvement in the global PSQI score of ≥ 3 points was considered a minimal clinically important difference. In our study, In the NNG group the score goes from 20.6 to 10.5; in TEG from 20.1 to 11.2; and in CG from 20.8 to 20.6. In ESS scale, It is estimated  [59] that the minimum clinically important improvement in the ESS lies between −2 and −3. Taking this variance in ESS into consideration it is time to reconsider the MCID to be between −5 and −6.  In our study, NNG from 15.2 to 6.40; in TEG from 14.7 to 10.1; in GC from 15.4 to 16.5. On the cognitive scale: one study [60] involving a survey of neurologists and geriatricians reported a mean MCID for the scale of 3.75 (95% confidence interval: 3.5-3.95).In our study, NNG from 22.7 to 30.7; TEG from 23.9 to 27.5 and GC from 18.6 to 18.3.
These data have been added to the "discussion" section: (page 14, line 300): “In the study by Cao, S et al (58) an improvement in the overall PSQI score of ≥ 3 points was considered a minimal clinically important difference. In our study, in the NNG group the score goes from a score of 20.6 to 10.5; in TEG from 20.1 to 11.2; and in CG from 20.8 to 20.6.”; (page 14, line 308): On the ESS scale, the minimum clinically important improvement in ESS is estimated (59) to be between -2 and -3. Given this variance in ESS, it is time to reconsider the MCID to be between -5 and -6.  In our study, NNG goes from a score of 15.2 to 6.40; in TEG from 14.7 to 10.1; in CG from 15.4 to 16.5. ; (page 15, line 317) : In addition, the study by J.S. Andrews et al.(60) involving a survey of neurologists and geriatricians reported a mean MCID for the scale of 3.75 (95% confidence interval: 3.5-3.95), in our study NNG from a score of 22.7 to 30.7, TEG from 23.9 to 27.5 and CG from 18.6 to 18.3.
58. Cao, S., Xin, B., Yu, Y. et al. Improvement of sleep quality in isolated metastatic patients with spinal cord compression after surgery. World J Surg Onc 21, 11 (2023). https://doi.org/10.1186/s12957-023-02895-0
59. Hunasikatti M. Low repeatability of the Epworth Sleepiness Scale and the need to redefine the minimal clinically important difference. J Clin Sleep Med. 2020;16(10):1827.
60. J.S. Andrews et al. / Alzheimer’s & Dementia: Translational Research & Clinical Interventions 5 (2019) 354-363
7. There is a misprint on page 4. Murray Johns developed the ESS, not J.M. Murray.
Response: Thank you for pointing this out. The name has been corrected: Page 5, line 180: “As the Murray Johns et al.”